# A Novel Defect Detection Method for Overhead Ground Wire

**DOI:** 10.3390/s24010192

**Published:** 2023-12-28

**Authors:** Yao Xiao, Lan Xiong, Zhanlong Zhang, Yihua Dan

**Affiliations:** 1School of Electrical Engineering, Chongqing University, Chongqing 400044, China; 202111131249@stu.cqu.edu.cn (Y.X.); lxiong@cqu.edu.cn (L.X.); 2Department of Electronic Engineering, Tsinghua University, Beijing 100084, China; microdanyihua@outlook.com

**Keywords:** overhead ground wire, defect, magnetic flux leakage signal, detection device

## Abstract

Overhead ground wires typically have strong axial tension and are prone to structural defects caused by corrosion and lightning strikes, which could lead to serious safety hazards. Therefore, it is important to detect defects accurately and quickly to avoid those problems. Existing defect detection methods for overhead ground wires are mainly traditional metal defect detection methods, including eddy current detection, ultrasonic detection, and manual visual inspection. However, those methods have problems of low detection efficiency, high environmental requirements, and insufficient reliability. To solve the above problems, this paper studies a novel type of defect detection technology for overhead ground wire. Firstly, the magnetic leakage characteristics around the defects of overhead ground wires are analyzed, and the defect detection device is designed. Then, the influence of air gap, lift-off distance, defect width, and cross-sectional loss rate on the magnetic flux leakage signal is studied, a novel defect detection method for overhead ground wire is proposed, and experimental verification is carried out. The results show that the proposed method can accurately locate and quantify the defect, which has the advantages of good reliability and high efficiency and lays the foundation for preventing accidents caused by defective overhead ground wires.

## 1. Introduction

The overhead ground wire is mainly used to reduce lightning overvoltage and sometimes for data transmission, which is an important part of the power system. Due to the complex and variable operating environment of overhead ground wire, the probability of a defect in the overhead ground wire structure is extremely high [1,2,3]. The axial tension of the overhead ground wire structure will have a large stress concentration at the defects’ places. If the defects cannot be found and repaired in time, it will eventually lead to the fracture and fall of the overhead ground wire, resulting in great economic loss and serious electric accidents [4,5,6].

In general, the existing detection methods for overhead ground wires still mainly rely on manual visual inspection methods [7]. This method has problems such as high randomness, low efficiency, etc. Overhead ground wire is a ferromagnetic material. The current detection methods for defects in ferromagnetic materials mainly include magnetic particle flaw detection, eddy current detection, ultrasonic detection, X-ray detection, thermal imaging detection, magnetic leakage detection, and so on [8]. Magnetic particle flaw detection technology is commonly used in the crack detection of aviation products, trains, and ships. The magnetic powder sprayed on the ferromagnetic material will gather in the defective part, and then the cracks are identified using manual inspection or machine vision [9]. The method has high requirements for both the environment and the inspector’s skills. The eddy current detection method determines the characteristics of the defects by analyzing the magnetic field changes caused by the eddy currents. Eddy current detection has a low detection depth and is susceptible to environments, which makes it difficult to realize higher precision measurements [10,11,12]. Magnetic particle and eddy current detection can only detect defects near the surface of the overhead ground wire; ultrasonic detection methods can effectively detect internal defects in the metal structure. Ultrasonic detection methods determine the defects by analyzing ultrasonic waves reflected in defective areas [13,14]. However, ultrasonic detection requires surface treatment, and the signal is also prone to interference, reducing the accuracy of the detection results. X-ray detection technology detects the object by light energy irradiation and penetration of the object, but the hazards of radiation to the human body limit the wide application of this technology [15]. The thermal imaging detection method determines the faults of electric equipment and ferromagnetic materials by detecting abnormal heating. Induced leakage current would heat the defective place of the overhead ground wire, and the thermal image of the overhead ground wire can be collected to judge the defective situation [16]. The reliability of this method is low due to the small abnormal temperature rise at the defects of the overhead ground wire. In summary, the existing manual visual inspection method has problems such as high randomness and subjectivity. Most of the existing methods for detecting defects in ferromagnetic materials have problems such as low detection efficiency, low precision, high detection requirements, lack of reliability, radiation risk, etc. Therefore, a scientific and standardized defect detection method for overhead ground wires is important for the power sector.

At present, the leakage magnetic detection method for steel wire ropes presents the advantages of high efficiency and reliability, and the overhead ground wire and steel wire ropes have similar structural characteristics and material properties, so the leakage magnetic detection method has a huge potential in the field of defect detection of overhead ground wire [17,18]. Refs. [19,20] studied the influence of magnet size, magnetic pole spacing, and lift-off distance on the magnetic leakage signal by the element simulation method. A simple magnetic leakage detection device was designed. However, the recognition accuracy of the device needs to be improved. To accurately and quantitatively detect the degree of wire rope damage, Ref. [21] selects a higher precision and a greater number of sensors. In terms of algorithms, Refs. [22,23,24] introduced wavelet transform, deep learning, convolution, and other algorithms into the quantitative detection of wire rope damage and achieved certain results, but still cannot fully meet the quantitative detection requirements of wire rope. To solve the current problems in the detection of defects in overhead ground wires, this paper studies the defect detection technology of overhead ground wires based on the principle of magnetic leakage detection. This paper mainly includes the design of the device, the analysis of the forward calculation, and the inverse calculation of the size of the defects. Firstly, the principle of magnetic leakage detection is analyzed, and a device suitable for defect detection for overhead ground wire is designed. Then, the influencing factors of magnetic leakage defect detection for overhead ground wire are analyzed based on the finite element model, and the change rules of defect size and magnetic signal are studied. Finally, the defect detection method for overhead ground wire is proposed and verified by experiments.

## 2. Defect Detection Theory of Overhead Ground Wire

### 2.1. The Basic Principle of Detecting the Defect of Overhead Ground Wire

The magnetic flux leakage defect detection principle for overhead ground wires is depicted in Figure 1. A permanent magnet, yoke, air gap, and the overhead ground wire form a magnetic flux circuit, which magnetizes the overhead ground wire axially. When the overhead ground wire is not damaged, the magnetic flux lines are mostly constrained within the wire, and the background magnetic field in the air is very weak. However, when a defect occurs in the overhead ground wire, there is an increase in magnetic resistance at the defect site, resulting in distortion in the magnetic field. The original magnetic flux lines cannot entirely pass through the wire at the defect location, causing some of the magnetic flux to leak from the defect area and form a magnetic leakage field. Figure 2a and Figure 2b, respectively, show the distribution of the magnetic field axial component *B_x_* and the magnetic field radial component *B_z_* along the dotted line in Figure 1b. It can be seen from the figure that the magnetic leakage field signal has obvious distortion at the defect: *B_x_* has a maximum value at the defect. The maximum and minimum values of *B_z_* appear at both ends of the defect, and the value of *B_z_* crosses zero at the center of the defect. The characteristics of magnetic leakage field signals *B_x_* and *B_z_* are closely related to the defect size. The characteristics of the magnetic leakage field are extracted by magnetic sensor measurement, and combined with the signal inversion method, the defect size of overhead ground wire can be accurately detected.

### 2.2. Finite Element Calculation Theory of Leakage Magnetic of Overhead Ground Wire

Based on Maxwell’s system of equations, the leakage magnetic field distribution of the overhead ground wire in Figure 1 can be described by Equations (1)–(4):(1)∇×H=J
(2)∇·B=0
(3)$$B=\mu_{0}\mu_{r}H$$
(4)B=∇×A
where *H* is the magnetic field strength vector; *J* is the current density vector; *B* is the magnetic flux density vector; *μ*_0_ is the vacuum permeability; *μ*_r_ is the relative permeability of the overhead ground wire; *A* is the vector magnetic potential.

Equation (5) is obtained from Equations (1)–(4):(5)$$\nabla \times \left(\frac{1}{\mu_{r}\mu_{0}}\nabla \times A\right)=J$$

Then, the solution region and boundary conditions are set, and Equation (5) is solved numerically by iterative finite element analysis. Finally, the leakage magnetic field distribution of the defective overhead ground wire is calculated.

## 3. Overhead Ground Wire Defect Detection Device Design

Design the magnetic leakage defect detection device for overhead ground wire, as shown in Figure 3 below. The workflow of the overhead ground wire magnetic leakage detection system is depicted in Figure 4. The functions implemented by the system are modularized, and the main modules in the hardware circuit include the microcontroller control module, AD acquisition module, data storage module, signal conditioning module, wireless transmission module, and more.

The design of an excitation device holds a crucial position in overhead ground wire defect detection. The saturation magnetization of the overhead ground wire is a prerequisite for accurate defect detection, which can significantly affect the precision of the detection process. Magnetizing the overhead ground wire to saturation results in leakage magnetic signals that are unaffected by material properties, providing stability and accuracy. To reduce the cost of the detection device and ensure its compact structure for portability, the integration of sensors and the excitation structure into a unit has been employed. The permanent magnet NdFeB52 is used to locally axially magnetize the overhead ground wire, and the yoke is used as the magnetic conducting path, further enhancing the magnetization effect. Apart from the excitation device, the air gap between the permanent magnet and the surface of the overhead ground wire, as well as the lift-off distance between the sensor and the surface of the overhead ground wire, are critical factors affecting the accuracy of the detection. This device allows for independent adjustment of these factors through a screw-adjustable height mechanism, optimizing the measurement performance.

The selection of magnetic-sensitive components is the key point of sensor probe design. Hall elements and coils are commonly used in magnetic flux leakage detection. Coils have larger physical dimensions, which are not conducive to compact device design, and they have slower response times. Therefore, the Hall sensor model HG-166A from AsahiKASEI Corporation (Tokyo, Japan) is chosen as the magnetic field sensor. This sensor is characterized by its compact size, rapid response, low power consumption, and high reliability. In practical measurements, it is necessary to simultaneously measure two magnetic field components in two different directions, each with a magnitude of several hundred Gauss. Three-axis Hall sensors can fulfill the requirement of simultaneously detecting two directions of the magnetic field but may not meet the full measurement range requirements. Therefore, two Hall sensors are arranged vertically to measure the magnetic field components in two directions simultaneously. The voltage output range of the Hall sensors is ±2.5 V. During detection, the device’s movement speed is relatively low, and the sampling rate of the analog-to-digital converter (ADC) is not stringent. Hence, the measurement requirements are met by employing the ADS1262 analog-to-digital converter from TEXAS INSTRUMENTS (Dallas, TX, USA). The entire system’s data acquisition, storage, and transmission functions are controlled by the STM32F103RCT6 microcontroller (Geneva, Switzerland).

## 4. Analysis of Magnetic Flux Leakage Distribution Characteristics

The air gap and lift-off distance have a great influence on the magnetic leakage signal. To ensure the stability and reliability of the measurement signal, it is necessary to set a reasonable air gap and lift-off distance before measurement. Therefore, a finite element model is established to analyze the influence of air gap and lift-off distance on the magnetization degree and magnetic leakage signal of the overhead ground wire and to determine the reasonable setting range of air gap and lift-off distance. Then, analyze the change rule of defect size and magnetic leakage signal, and propose the detection method of defect size of overhead ground wire.

### 4.1. Finite Element Modeling for Magnetic Leakage Analysis of Overhead Ground Wire

References [25,26] indicate that, considering factors such as geometric modeling, mesh partitioning, and computation time, and given that the multi-strand structure of overhead wire has a negligible impact on the leakage magnetic signal at defect locations, a finite element model can be established using a smooth cylindrical equivalent representation for practical overhead wire. Therefore, this paper establishes a finite element model for magnetic leakage analysis of overhead ground wire, as shown in Figure 5 below. Where *δ* represents the air gap, *f* represents the lift-off distance, which is the distance between the magnetic field sensor and the surface of the overhead ground wire. The diameter of the overhead ground wire is 9 mm, and the length is 1000 mm.

Due to the influence of different factors and environments, various types of defects, such as broken strands, abrasion, corrosion, fatigue, thermal damage from lightning strikes, etc., can occur on overhead ground wires. These defects have different manifestations, and they may occur simultaneously and interact with each other. Therefore, it is necessary to quantify the size of defects using a uniform standard. The reduced cross-sectional area is the direct cause of the decline in the bearing capacity of the overhead ground wire, so it is reasonable to use the defect width and cross-sectional loss rate to define the size of the defect. The schematic diagram of the defect is shown in Figure 6 below; the location of the defect is at *x* = 500 mm, and the cross-sectional area of the defect is bowed. *a* represents the width of the defect, *S*_1_ represents the area lost at the defect, and *S*_2_ represents the remaining area at the defect, and the cross-sectional loss rate of the overhead ground wire is *η_s_*, which is calculated by the following equation:(6)$$\eta_{s}=\frac{S_{1}}{S_{1}+S_{2}}\times 100\%$$

Based on the magnetic properties of the materials, appropriate constitutive relationships are chosen for different materials. The constitutive relationship for the air domain and ferrous materials is represented by the relative magnetic permeability, with values of 1 and 4000, respectively. The magnetic properties of the permanent magnet are expressed in terms of the remanent flux density, which has a value of 1.5 T. The constitutive relationship for the overhead ground wire utilizes a B-H curve, as shown in Figure 7.

After setting appropriate control equations and boundary conditions, calculations are performed using the Magnetic Field module within the AC/DC interface in COMSOL. Finally, the solution of the finite element model of the leakage magnetic field of the overhead ground wire is obtained.

The air gap *δ* affects the degree of magnetization of the overhead ground wire and, thus, the measurement signal. The lift-off distance *f*, defect width *a*, and cross-sectional loss rate *η_s_* also significantly affect the measurement signal. Therefore, the effects of air gap *δ*, lift-off distance *f*, defect width *a*, and cross-sectional loss rate *η_s_* on the measurement signal are investigated in Section 4.2 and Section 4.3, respectively.

### 4.2. Effect of Air Gap on Magnetic Leakage Signal

The air gap *δ* affects the degree of magnetization of the overhead ground wire. Uniform and saturated magnetization of defect-free overhead ground wires by the detection device is required for a stable and strong measurement signal. When *δ* = 5 mm, the magnetic field distribution across the cross-sectional area of the overhead ground wire without defect is shown in Figure 8. In this figure, it can be observed that two permanent magnets create a closed magnetic flux path, confining almost all magnetic field lines within the overhead ground wire. The magnetic field distribution in the magnetized region of the overhead ground wire is uniform, with a magnetic field strength of approximately 2 T. The region achieves magnetic saturation. Figure 9 shows the relationship between the magnetization level of the overhead ground wire and the air gap *δ*. Smaller air gaps result in higher magnetization levels. When the air gap increases from 5 mm to 30 mm, the magnetization level of the overhead ground wire decreases from 1.8 T to 1.55 T, a decrease of 0.25 T.

When the overhead ground wire has a defect (*a* = 5 mm, *η_s_* = 50%), its magnetic field distribution is shown in Figure 10. Magnetic field lines exhibit distortion at the defect site, with magnetic lines leaking into the air above the defect. Extract the value of magnetic leakage intensity *B* at 5 mm above the defect, and the change of *B* with the air gap is shown in Figure 11. Smaller air gaps result in higher leakage magnetic field signal strength. As the air gap increases from 5 mm to 30 mm, the intensity of the leakage magnetic signal above the overhead ground wire defect decreases from 380 Gs to 180 Gs, marking a decline of over 50%.

In summary, a smaller air gap leads to higher magnetization levels in the overhead ground wire, resulting in stronger leakage magnetic field signals. However, when the gap is too small, the magnetic force between the device and the overhead ground wire is significant, making it less convenient for the device to move along the wire’s surface. Therefore, to ensure measurement accuracy and facilitate the movement of the device, the appropriate range for the air gap *δ* is 5~10 mm.

### 4.3. Effect of Lift-Off Distance on Magnetic Leakage Signal

The lift-off distance *f* refers to the distance between the sensor and the surface of the overhead ground wire. To detect the leakage magnetic signal at the defect site effectively, *f* must be within a certain range; exceeding this range would result in the signal being overshadowed by the background magnetic field. If *f* is too small, the leakage magnetic signal becomes unstable, making it crucial to select an appropriate lift-off distance.

A vertical spatial magnetic field observation line, as shown in Figure 12, is established directly above the defect, perpendicular to the surface of the overhead ground wire. Magnetic field values are extracted along this observation line under both defect-free and defective conditions, as shown in Figure 13. The horizontal axis, *z*, ranges from 0 mm (near the surface) to 40 mm (the other end of the observation line). When there is no defect, the magnetic field values along the observation line remain nearly constant, at approximately 100 g. However, when there is a defect, the magnetic field values along the observation line combine with the leakage magnetic field emanating from the defect, resulting in changes in the magnetic field along the observation line. The magnetic field values closer to the surface of the overhead ground wire significantly increase, with the magnitude growing as the observation point moves closer to the wire’s surface. Conversely, the magnetic field values further from the surface remain almost the same as the original field, showing minimal variation. To capture the leakage signal containing defect information effectively, the sensor should be positioned within the range of 0 < *f* < 20 mm. However, when *f* < 5 mm, the gradient of the magnetic field signal changes substantially, leading to signal instability and reduced noise immunity.

### 4.4. Effect of Defect Size on Magnetic Leakage Signal

With an air gap of *δ* = 5 mm and a lift-off distance of *f* = 5 mm in the excitation structure, this paper studies the relationship between the overhead ground wire defects and the leakage magnetic signals. Figure 14a shows the distribution characteristics of the magnetic field signal *B_x_* measured by the Hall sensor when the defect width is *a* = 10 mm. *B_x_* exhibits peak-like distribution characteristics at the defect site, with maximum *B_x_*_-max_ appearing at the center of the defect. The value of *B_x_*_-max_ increases with the increase in cross-sectional loss. The value of *B_x_*_-max_ presents a monotonic relationship with the magnitude of cross-sectional loss. By conducting several simulations, the distribution characteristics of the detection signal at the defect site were obtained for a range of defect widths from 0.5 mm to 60 mm and cross-sectional loss rate *η_s_* from 1% to 90%. *B_x_*_-max_ values were extracted, and the relationship between *B_x_*_-max_, defect width *a*, and cross-sectional loss rate *η_s_* is shown in Figure 14b. It can be seen that both defect width and cross-sectional loss rate will significantly influence the magnitude of *B_x_*_-max_. Figure 14c shows the relationship between defect width and *B_x_*_-max_. When the defect width is less than 5 mm, *B_x_*_-max_ increases with an increase in defect width. When the defect width exceeds 5 mm, *B_x_*_-max_ decreases with a further increase in defect width. Figure 14d shows the relationship between the cross-sectional loss rate and *B_x_*_-max_. As the cross-sectional loss rate increases, *B_x_*_-max_ also increases. However, when the cross-sectional loss rate is less than 20%, the change in *B_x_*_-max_ is not substantial.

Figure 15a shows the distribution characteristics of *B_z_* at the defect site when the defect width is 10 mm. When the cross-sectional loss rate *η_s_* is less than 20%, *B_z_* at the defect site remains relatively constant. However, when the *η_s_* exceeds 20%, *B_z_* exhibits distinctive peak-valley features, with *B_x_*_-max_ increasing as the cross-sectional loss of the overhead ground wire increases. The *x*-direction spacing between peaks and valleys is denoted as *L-B_z_,* and it is almost unaffected by the cross-sectional loss. Simulation results provide *B_z_* distribution characteristics for various defect widths. When the defect width is less than 2 mm, a *B_z_* does not display peak-valley features. For defect widths greater than or equal to 2 mm, statistical analysis of the relationship between *L-B_z_* and *η_s_* is shown in Figure 15b. The *L-B_z_* increases with increasing defect width. When the defect width is fixed, variations in *L-B_z_* are minimal. Consequently, the value of *L-B_z_* exhibits a strong correlation with the defect width.

In Section 4, through the design of three simulation conditions, the impact of air gap *δ*, lift-off distance *f*, defect width *a*, and cross-sectional loss rate *η_s_* on measurement signals is analyzed. This study reveals that smaller air gaps lead to higher magnetization of the overhead ground wire and stronger leakage of magnetic signals. However, very small distances result in increased magnetic forces between the device and the overhead ground wire, making it less convenient for the device to move along the wire’s surface. Excessive lift-off distance *f* can result in the masking of leakage magnetic signals, and overly small *f* values may yield unstable signals. The assessment of overhead ground wire defect width and cross-sectional loss rate can be based on comprehensive considerations of *L-B_z_*, *B_x_*_-max_, and *B_z_*_-max_.

## 5. Defect Detection Method for Overhead Ground Wire

Section 4 shows that the peak-to-valley horizontal spacing *L-B_z_* of *B_z_*, the peak *B_x_*_-max_ of *B_x_*, and the peak *B_z_*_-max_ of *B_z_* can be used as parameters for quantitatively determining the size of the defects, but the air gap and the lift-off distance affect the *B_x_* and *B_z_* signals. Once the overall device structure is determined, the magnet’s dimensions, magnetic strength, magnetic pole spacing, and other structural parameters remain fixed. Before measurement, it is necessary to optimize the magnetic flux leakage detection effect by adjusting the air gap *δ* and lift-off distance *f*. The air gap closely correlates with the magnetization level of the overhead ground wire, favoring a smaller air gap to ensure saturation magnetization. As the distance between the sensor and the overhead ground wire surface increases, the leakage magnetic signal rapidly decreases. Beyond a certain distance, the signal becomes exceedingly weak and is submerged within the background magnetic field, so that the magnetic leakage characteristic signal cannot be detected. Hence, the lift-off distance should not be too large. However, to ensure stable measurement signals, it should not be too small either, necessitating the determination of an appropriate lift-off distance through specific methods. Due to the nonlinear relationship and numerous features between the magnetic leakage characteristic signal and defect size, it is difficult to use fitting functions for defect size inversion. Therefore, following the collection of overhead ground wire magnetic leakage data, a neural network method is employed to perform defect size inversion. Based on the aforementioned analysis, a defect detection method for overhead ground wire is proposed, comprising two main parts: (a) Device design and determination of key structural parameters. (b) Extraction of characteristic parameters and defect size inference

The specific method steps are as follows:•Step 1: Device design and determination of key structural parameters. The designed structure of the overhead ground wire magnetic flux leakage detection device is shown in Figure 3. Finite element analysis is used to determine a suitable air gap and lift-off distance. Initially, a finite element analysis model is constructed to calculate the relationship between overhead ground wire magnetization intensity *B*_1_ and air gap *δ* as *B*_1_ = F_1_(*δ*). When *δ* > *δ*_0_, the overhead ground wire has not reached magnetic saturation, whereas when *δ* ≤ *δ*_0_, the wire has achieved magnetic saturation. Therefore, it is necessary for the air gap *δ* to be less than or equal to *δ*_0_ to achieve magnetic saturation in the overhead ground wire. Similarly, the relationship between leakage magnetic signal *B*_2_ and lift-off distance *f* is expressed as *B*_2_ = F_2_(*f*). This relationship between the magnetic leakage signal and lift-off distance is shown in Figure 13. If *f* = *a* and |∂*B*_2_/∂*f*| = 1, then the lift-off distance *f* is approximately equal to *a*.•Step 2: Magnetic leakage signal acquisition and preprocessing. The overhead ground wire leakage magnetic detection device scans along the wire, with magnetic field sensors collecting data at evenly spaced intervals along the wire’s axis. Unlike ideal signals obtained through finite element calculations, practical leakage magnetic signals contain considerable high-frequency noise and data anomalies. Therefore, preprocessing is essential before quantitative analysis. The signal processing workflow, as shown in Figure 16, includes the removal of outlier values using median filtering and the elimination of high-frequency noise using wavelet thresholding.•Step 3: Extraction of characteristic parameters. The quantitative identification of defect sizes in overhead ground wires is closely related to the extraction of characteristic parameters from the magnetic leakage signals. Therefore, the extraction of characteristics is a crucial step in the analysis of magnetic leakage signals. Based on the rule of how magnetic leakage signals change with defect size, this paper defines the following three magnetic leakage parameters:1.The peak value of the magnetic leakage signal *B_x_* is denoted as *X*_1_. It represents the maximum value of the magnetic field component *B_x_* at the location of the defect.2.The peak value of the leakage signal *B_z_* is denoted as *X*_2_. It represents the maximum value of the magnetic field component *B_z_* at the location of the defect.3.The peak-to-valley horizontal spacing of the magnetic leakage signal *B_z_* is denoted as *X*_3_. It is defined as the distance between the peak and valley of the magnetic leakage signal *B_z_* in the horizontal direction.•Step 4: Constructing a database of samples. After defining the above characteristic parameters, the magnetic leakage signal on the defective surface of the overhead ground wire can be obtained by using the magnetic dipole computational model, the finite element computational model, and experiments. Then, characteristic values can be extracted to construct a database of magnetic leakage detection samples.•Step 5: GA-BP (genetic algorithm-back propagation) neural network to predict defect size GA-BP (genetic algorithm-back propagation) neural network to predict defect size. The BP (back propagation) neural network has strong self-adaptation and self-learning abilities and can process large-scale data in parallel. However, the search direction of the BP neural network is relatively simple, which makes the network easy to fall into the local optimal state. GA (genetic algorithm) is an adaptive parallel optimization algorithm. It simulates the mechanisms of biological genetics and evolution and has the advantages of global search, high parallelism, and strong generalization ability. The GA-BP neural network uses GA to optimize the initial weights and thresholds of the neural network, which are located near the global optimum. Then, the initial weights and thresholds of the neural network are given to the neural network for network training and learning until it converges to find the global optimal solution. The schematic diagram of the BP neural network structure and the specific algorithm flow of the GA-BP neural network are shown in Figure 17 and Figure 18.

The process of predicting the size of defects in overhead ground wires using the GA-BP neural network is as follows:4.The sample data are divided into training and test data and normalized. Then, the number of nodes in the input, hidden, and output layers of the BP neural network is determined.5.Optimize the BP neural network using the genetic algorithm toolbox (GAOT) of Matlab. Save the BP neural network model when the error in the training data meets the objective.6.Test the network model using test data. Construct a judging metric to judge the prediction accuracy of this network and save the neural network models that predict well.7.The magnetic leakage signal of the overhead ground wire to be tested is obtained by using the magnetic leakage detection device, and the magnetic leakage characteristic parameters are extracted and brought into the trained neural network model in (c) to obtain the defect size of the overhead ground wire.

## 6. Experimental Verification

To construct a sample database of real magnetic leakage signals, a simulation experiment platform is built, as shown in Figure 19 below. The dimensional parameters of the device structure are the same as those of the simulation model shown in Figure 3. According to the simulation analysis in Section 2, the air gap *δ* = 5 mm and the lift-off distance *f* = 5 mm. The overhead ground wire of model GJ-50 is taken as the object to be detected, the stranded structure is ignored, and the overhead ground wire is simplified as a cylindrical mild steel bar with a diameter of 9 mm. The sensor is a linear Hall sensor, and the magnetic field measurement range is ±500 Gs. The model of the acquisition card is AD7606, 16-bit resolution, with the highest sampling rate of 200 Ksps. The collected data are transmitted to the upper computer through the data line. The telescopic shaft controls the detection scanning speed through unidirectional expansion and contraction, and the speed can be adjusted from 4 mm/s to 30 mm/s. The role of the controller is to control the expansion and contraction of the telescopic axis.

Using the wire-cut processing method, 96 samples of overhead ground defect models were produced on simplified overhead ground wires, as shown in Table 1 below.

The magnetic leakage signal data for each sample were obtained, and the signals were preprocessed by the median filtering method and the wavelet threshold denoising method. Subsequently, the extracted leakage magnetic feature data were used to construct the sample database. The physical diagrams of defects and the waveforms of magnetic leakage signals of samples 11, 28, 45, and 62 are shown in Figure 20 below. The feature data of 78 sets of samples were randomly selected as the training set, while the feature data of the remaining 18 sets of samples were used as the test set.

The GA-BP neural network was constructed with Matlab, and the learning model error of the neural network under different learning rates and different training times was compared. The learning rate was set to 0.1, and the training times were set to 100. A 3-layer BP neural network structure is established. The input layer has 3 nodes (corresponding to 3 features X_1_~X_3_), the hidden layer is set to 7 nodes, and the output layer has 2 nodes (corresponding to defect width and defect cross-sectional loss rate). The selection of the hidden layer function and output layer function has a great influence on the prediction accuracy of the BP neural network. In this paper, the hidden layer function is the logsig function, and the output layer function is the transig function.

After several training sessions, the values of the genetic algorithm and BP neural network parameters were determined, as shown in Table 2 and Table 3.

Root Mean Square Error (RMSE) and Mean Absolute Percentage Error (MAPE) were used to evaluate the inversion accuracy of the model, and the smaller the values of RMES and MAPE of the model, the higher the accuracy of the model inversion. The formulae for RMES and MAPE are shown in Equations (7) and (8).
(7)$$\mathrm{RMSE}=\sqrt{\frac{\sum_{i=1}^{n}{\left(x_{i}-x_{i}^{*}\right)}^{2}}{n}}$$
(8)$$\mathrm{MAPE}=\frac{100\%}{n}\sum_{i=1}^{n}\left|\frac{x_{i}-x_{i}^{*}}{x_{i}^{*}}\right|$$
where *x_i_* denotes the real value, *x_i_*^*^ denotes the model predicted value, *i* is the data number, and *n* is the total number of data points.

Figure 21 shows the predictive results of the GA-BP neural network for the defect width and cross-sectional loss rate of the overhead wire. As shown in Figure 21a,b, the RMSE for defect width prediction in the training set is 0.36693 with a MAPE of 1.1554%, and in the testing set, it is 0.42893 with a MAPE of 2.8281%. As shown in Figure 21c,d, the RMSE for defect cross-sectional loss rate prediction in the training set is 0.015747 with a MAPE of 4.2752%, and in the testing set, it is 0.052095 with a MAPE of 7.271%. The true values of defect width and cross-sectional loss rate are very close to the predicted values, indicating an overall good predictive performance. However, the predictive performance for defect width is superior to that for cross-sectional loss rate. The absolute error data for each sample in the testing set are presented in Table 4. The absolute errors of the predicted value of overhead ground wire width are all within 1 mm, and the absolute errors of the predicted value of overhead ground wire cross-sectional loss rate are within 3% for more than 80% of the samples. The potential reasons for cross-sectional loss rate prediction errors exceeding 3% may stem from vibrations during the measurement process and fitting errors within the neural network model itself. In conclusion, the error results and recognition accuracy meet the practical requirements of engineering applications.

## 7. Conclusions

Overhead ground wires operate in complex environments, suffer long-term corrosion, and are subjected to large axial tensions. Overhead ground wires are prone to structural defects, and further development can lead to serious power safety accidents. Accurate detection of the size of defects in the overhead ground wire can effectively avoid problems caused by structural damage to the overhead ground wire, so accurate detection of the size of defects in the overhead ground wire is essential to ensuring the safe and stable operation of transmission lines. To solve the problems of low detection efficiency, poor reliability, and high environmental requirements of the existing methods, this paper designs a new type of defect detection device for overhead ground wires and proposes a new type of defect detection method for overhead ground wires. The main conclusions are as follows:8.According to the influence of the air gap and lift-off distance on the magnetic leakage signal, the appropriate air gap range is determined to be 5~10 mm, and the lift-off distance range is 5~20 mm. The smaller the air gap and lifting distance, the stronger the magnetic leakage signal. However, the air gap being too small will result in too much magnetic force, making it difficult to move the device; the lift-off distance being too small will lead to signal instability. Increasing the lift-off distance improves the stability of the signal, but the leakage signal will be gradually covered by the background magnetic field, and the sensitivity will be reduced.9.*B_x_*, *B_z_*, and *L-B_z_* are determined as three characteristic quantities reflecting the size of defects. The peak values of *B_x_* and *B_z_* gradually increase with the increase in the cross-sectional loss rate of defects, and *L-B_z_* gradually increases with the increase in the width of defects. These three characteristic quantities are closely related to the defect size and can be utilized to evaluate the defect width and the defect cross-sectional loss rate.10.A defect detection method for overhead ground wire is proposed, which is verified by the experimental tests. The experimental results show that the absolute errors of the predicted value of overhead ground wire width are all within 1 mm, and the absolute errors of the predicted value of overhead ground wire cross-sectional loss rate are within 3% for more than 80% of the samples.

In summary, the proposed method can solve the problems of existing methods and has the advantages of greater measurement depth, no detection blind spots, good environmental adaptability, high detection accuracy, stability, etc. This work can detect defects in overhead ground wires accurately and efficiently, which improves the efficiency of overhead ground wire maintenance and has great engineering significance.

## Figures and Tables

**Figure 1 sensors-24-00192-f001:**
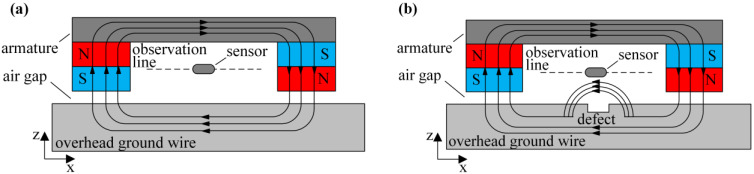
Overhead ground wire magnetic flux leakage detection schematic diagram: (**a**) Without defect; (**b**) Defective.

**Figure 2 sensors-24-00192-f002:**
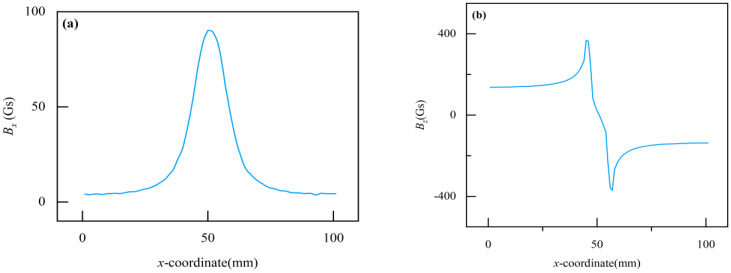
Magnetic flux leakage characteristic signal at the defect: (**a**) *B_x_*; (**b**) *B_z_*.

**Figure 3 sensors-24-00192-f003:**
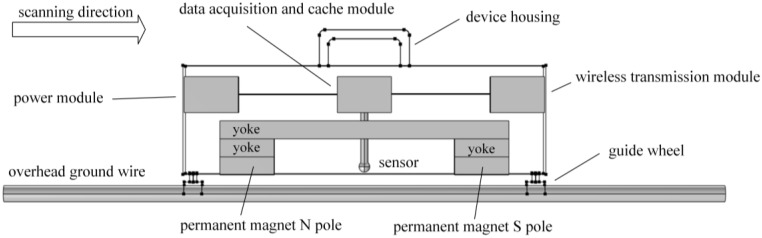
Structural design schematic diagram of the device.

**Figure 4 sensors-24-00192-f004:**
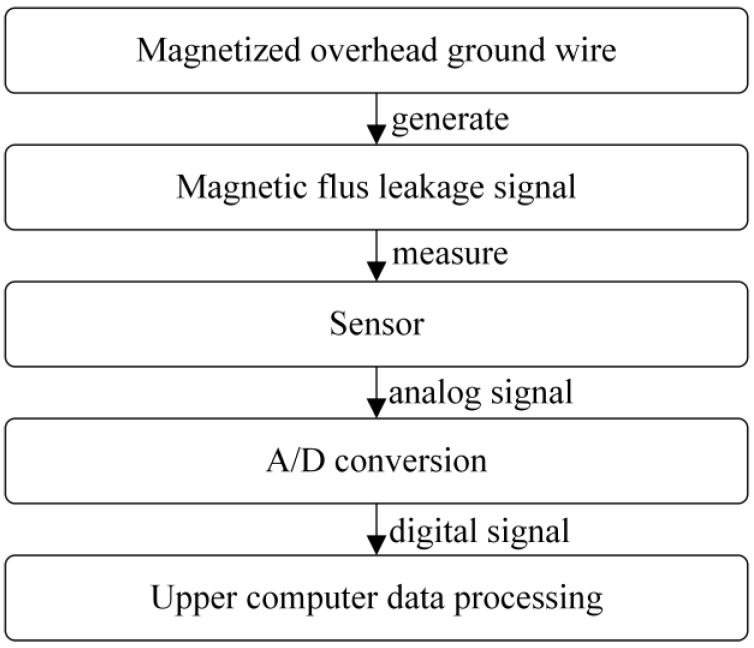
System workflow.

**Figure 5 sensors-24-00192-f005:**
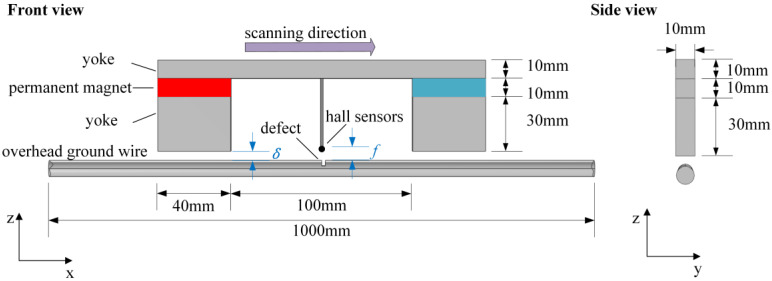
Overhead ground wire simulation calculation model and parameter diagram.

**Figure 6 sensors-24-00192-f006:**
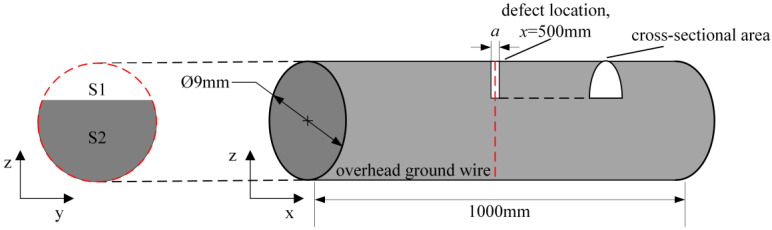
Overhead ground wire defect diagram.

**Figure 7 sensors-24-00192-f007:**
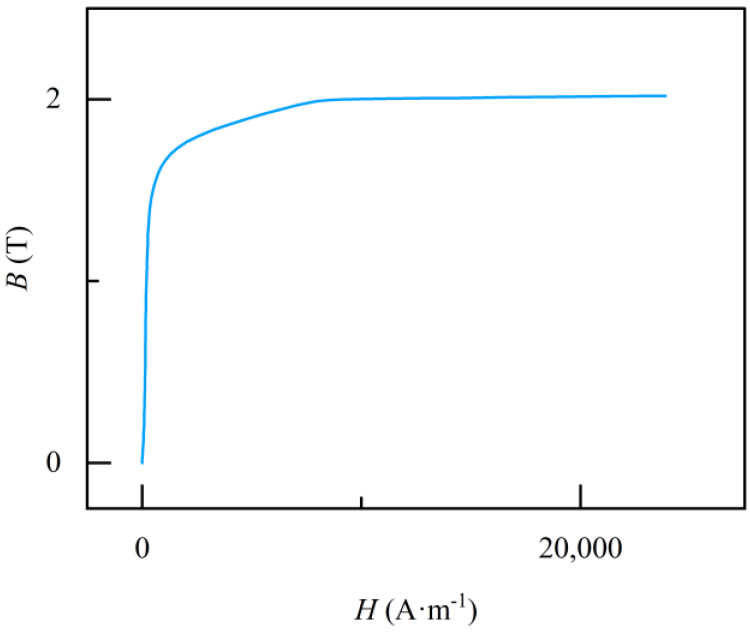
Overhead ground wire B−H curve.

**Figure 8 sensors-24-00192-f008:**
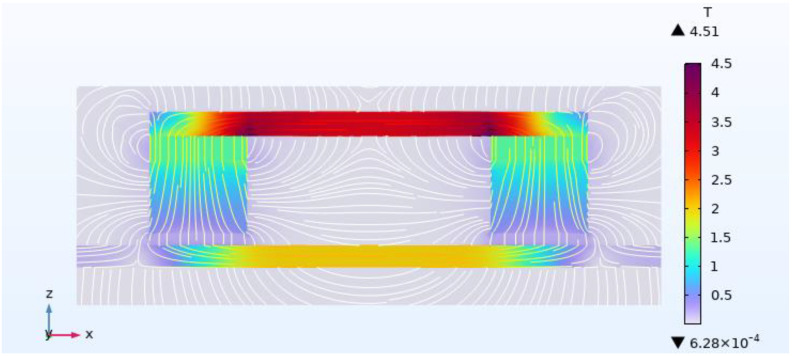
Magnetic field distribution cloud diagram of overhead ground wire without defects.

**Figure 9 sensors-24-00192-f009:**
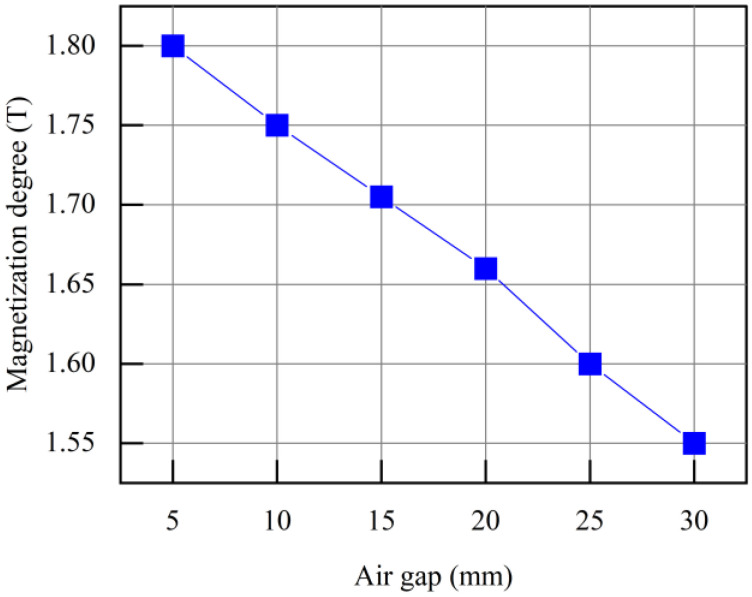
Relationship between the magnetization level of the overhead ground wire and the air gap.

**Figure 10 sensors-24-00192-f010:**
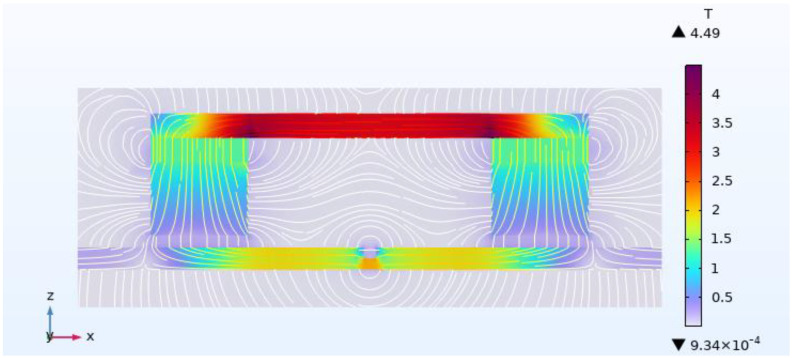
Magnetic field distribution cloud diagram of overhead ground wire with defects.

**Figure 11 sensors-24-00192-f011:**
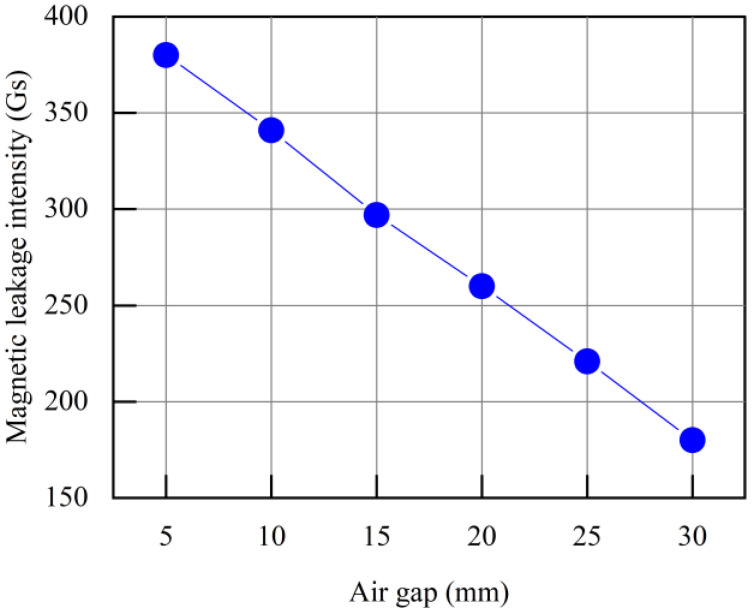
At 5 mm above the defect, there is a relationship between the magnetic flux leakage intensity and the air gap.

**Figure 12 sensors-24-00192-f012:**
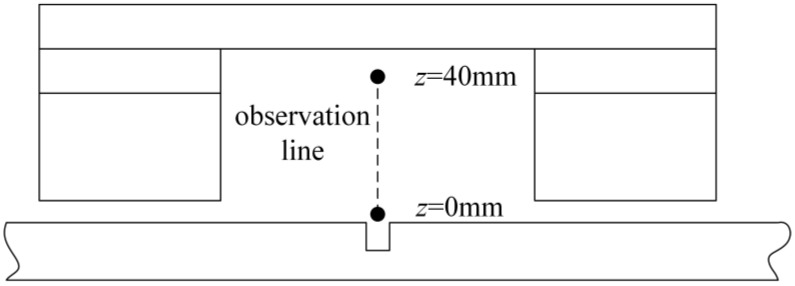
Observation line diagram.

**Figure 13 sensors-24-00192-f013:**
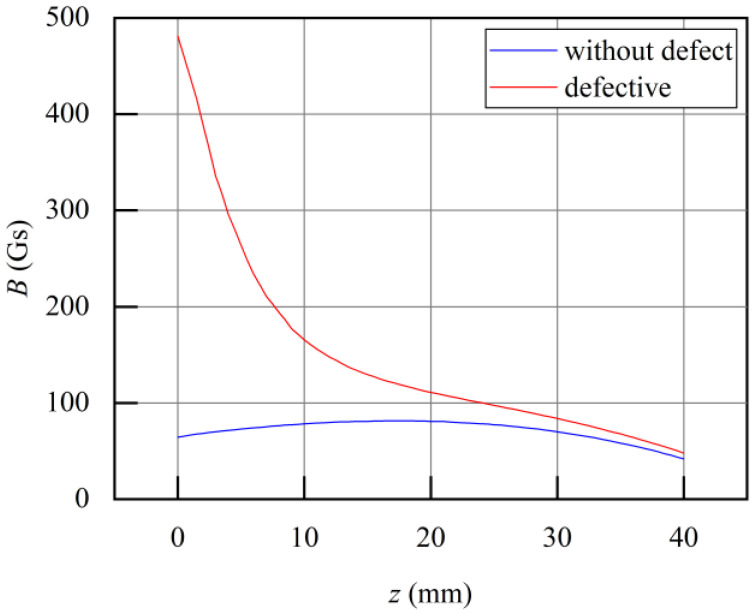
The magnetic field changes on observation lines with and without defects.

**Figure 14 sensors-24-00192-f014:**
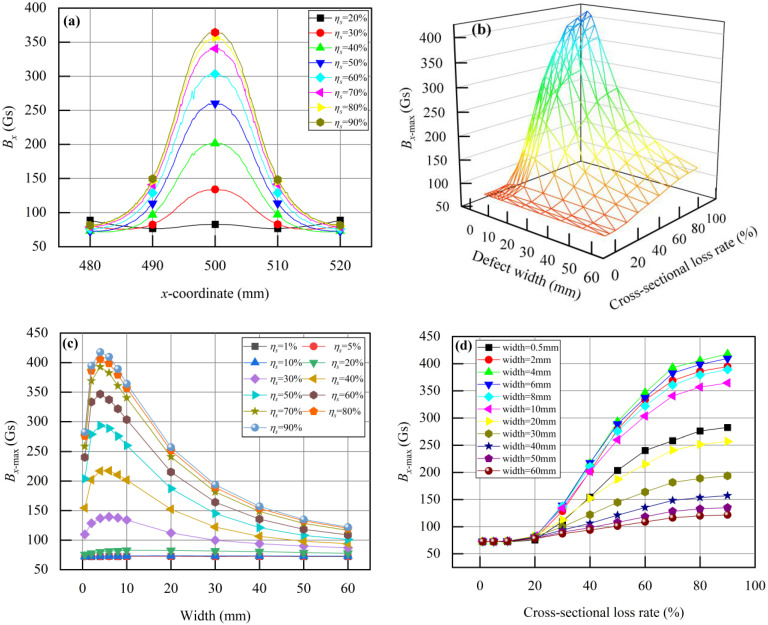
The relationship between defect width, cross-sectional loss rate, and detection signal *B_x_*: (**a**) When the defect width *a* = 10 mm and the cross-sectional loss rate *η_s_* = 20%~90%, the distribution characteristics of *B_x_*; (**b**) The relationship between *B_x_*_-max_ and defect width *a* = 0.5~60 mm, cross-sectional loss rate *η_s_* = 1%~90%; (**c**) The relationship between *B_x_*_-max_ and defect width *a* = 0.5~60 mm; (**d**) The relationship between *B_x_*_-max_ and cross-sectional loss rate *η_s_* = 1%~90%.

**Figure 15 sensors-24-00192-f015:**
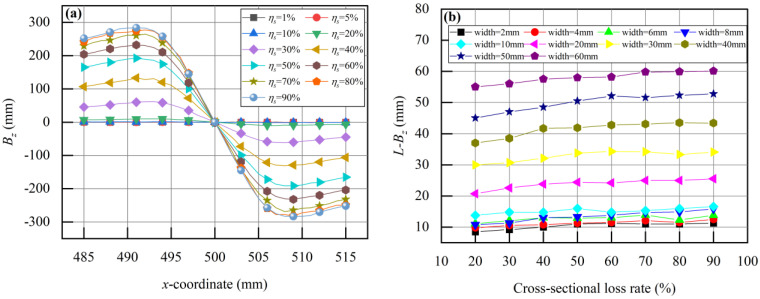
The relationship between defect width and detection signal *B_z_*: (**a**) The distribution characteristics of defect width *a* = 10 mm, *B_z_*; (**b**) The relationship between the peak-valley spacing *L-B_z_* and the cross-sectional loss rate with different defect widths (2 mm~60 mm).

**Figure 16 sensors-24-00192-f016:**
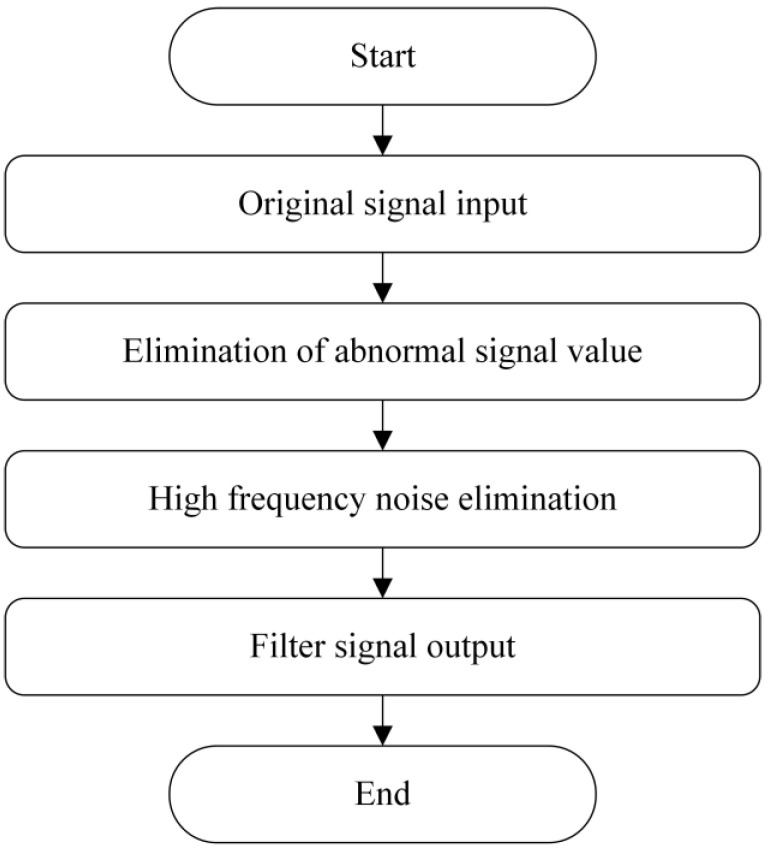
Signal processing flow.

**Figure 17 sensors-24-00192-f017:**
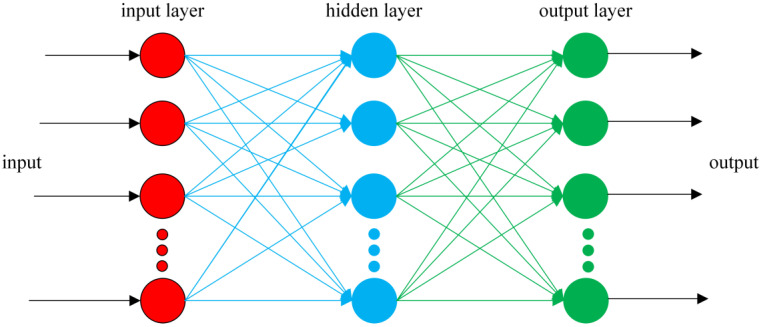
BP neural network structure diagram.

**Figure 18 sensors-24-00192-f018:**
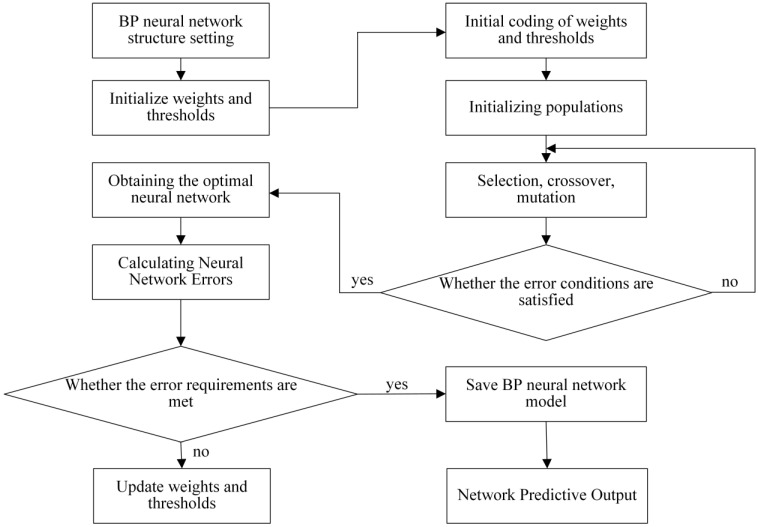
GA-BP neural network training process.

**Figure 19 sensors-24-00192-f019:**
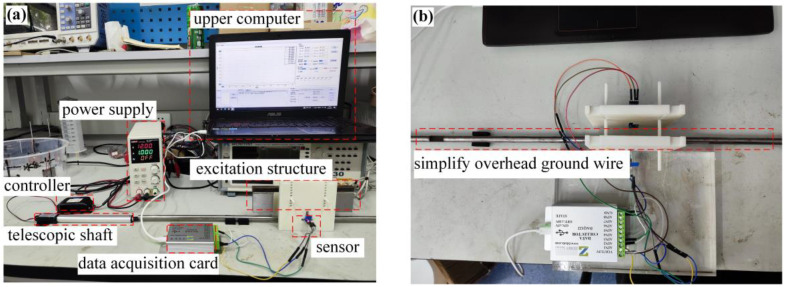
Simulation experiment platform: (**a**) Overall diagram; (**b**) Data acquisition diagram.

**Figure 20 sensors-24-00192-f020:**
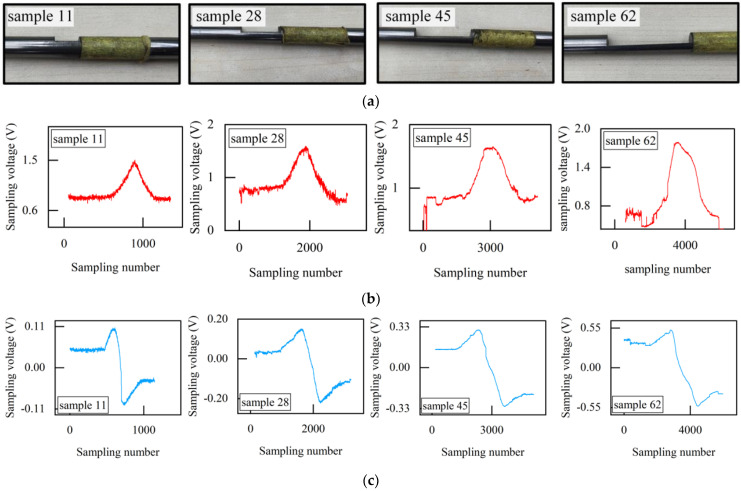
Physical diagram of the defect and waveforms of *B_x_* and *B_z_* leakage signals: (**a**) Physical diagrams; (**b**) *B_x_*; (**c**) *B_z_*.

**Figure 21 sensors-24-00192-f021:**
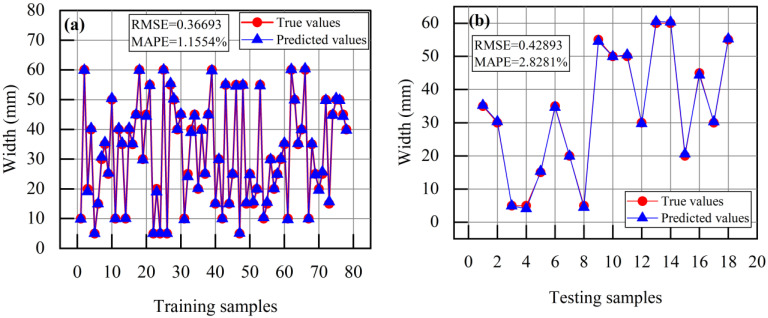
Prediction results of the GA-BP neural network: (**a**) Width prediction training set; (**b**) Width prediction testing set; (**c**) Cross-sectional loss rate prediction training set; (**d**) Cross-sectional loss rate prediction testing set.

**Table 1 sensors-24-00192-t001:** Overhead ground wire defect size parameters.

Sample Number	Cross-Sectional Loss *η_s_* (%)	Defect Width *a* (mm)	Sample Number	Cross-Sectional Loss *η_s_* (%)	Defect Width *a* (mm)
1	10	5	49	10	35
2	20	5	⁝	⁝	⁝
3	30	5	56	80	35
4	40	5			
5	50	5	57	10	40
6	60	5	⁝	⁝	⁝
7	70	5	64	80	40
8	80	5			
			65	10	45
9	10	10	⁝	⁝	⁝
⁝	⁝	⁝	72	80	45
16	80	10			
			73	10	50
17	10	15	⁝	⁝	⁝
⁝	⁝	⁝	80	80	50
24	80	15			
			81	10	55
25	10	20	⁝	⁝	⁝
⁝	⁝	⁝	88	80	55
32	80	20			
			89	10	60
33	10	25	⁝	⁝	⁝
⁝	⁝	⁝	96	80	60
40	80	25			
41	10	30			
⁝	⁝	⁝			
48	80	30			

**Table 2 sensors-24-00192-t002:** The parameter value of the genetic algorithm.

Population Size	Iteration Times	Variable Length	Crossover Probability	Mutation Probability
20	50	6	0.3	0.1

**Table 3 sensors-24-00192-t003:** BP neural network parameter values.

Hidden Layer Function	Output Function	Training Function	Frequency of Training	Rate of Learning	Number of the Hidden Layer Nodes
logsig	tansig	trainlm	100	0.2	9

**Table 4 sensors-24-00192-t004:** Overhead ground wire defect size prediction results.

Serial Number	True Values		Predicted Values		Absolute Error	
	Width (mm)	Cross-Sectional Loss (%)	Width (mm)	Cross-Sectional Loss (%)	Width (mm)	Cross-Sectional Loss (%)
1	35	70	35.26355	72.17175	0.263549	2.171749
2	30	20	30.35438	18.62692	0.354378	−1.37308
3	5	50	4.928669	32.72245	−0.07133	−17.2776
4	5	10	4.055256	11.57531	−0.94474	1.575307
5	15	50	15.31036	50.01758	0.31036	0.017577
6	35	80	34.64018	77.48767	−0.35982	−2.51233
7	20	50	19.93007	48.96714	−0.06993	−1.03286
8	5	60	4.50664	48.40352	−0.49336	−11.5965
9	55	80	54.55471	80.18	−0.44529	0.179998
10	50	80	49.93903	77.26825	−0.06097	−2.73175
11	50	30	50.43603	30.63613	0.436032	0.636135
12	30	70	29.70099	75.09386	−0.29901	5.093864
13	60	30	60.44312	31.54727	0.443117	1.54727
14	60	40	60.36944	38.92275	0.369444	−1.07725
15	20	40	20.60974	41.01483	0.609742	1.014834
16	45	70	44.34946	69.7722	−0.65054	−0.2278
17	30	60	30.24246	60.71333	0.24246	0.713328
18	55	50	55.27708	50.23921	0.277084	0.239212

## Data Availability

Data are contained within the article.

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
