# Peer review of "A Novel Defect Detection Method for Overhead Ground Wire"

_sensors, 2023, doi:10.3390/s24010192_

Round 1

Reviewer 1 Report

Comments and Suggestions for Authors

It is of great significance of this paper carrying out the defect detection research of overhead wires. However, the presentation of the magnetic flux leakage sensor used in this paper is not specific enough, resulting in the innovation and how to apply this sensor in the field is not very clear. Detailed commemts are as follows.

1. The overhead ground wire is very long and high, so how to installed the sensor device including the magnets onto the wire in field? 

How to ensure that the magnetization line just passes through the defect, because the magnets may rotate around the axis of the wire. Since the magnet will hold the wire tightly, how to smoothly mobile the devices? 

2. It is a traditional method to detect the pits and cracks of ferromagnetic materials by magnetization and magnetic flux leakage method. So what are the differences between the simulation/experimental studies in this paper and the previous studies, including the research methods and results?

3. Is the overhead ground wire single strand or multiple strands? Why does this paper use a single strand to do simulation and experimental research?

4. Magnetic flux leakage detection can be very sensitive and accurate to such large defects as listed in Table 1, why is neural network required? Is it because the device wasn't designed properly? The test samples of neural network cannot be sufficient in this paper.

Comments on the Quality of English Language

Minor editing of English language required.

Reviewer 2 Report

Comments and Suggestions for Authors

Interesting work that needs a little improvement:

1. The design information on lines 137-167 is limited, with no data containing trade names of the component and the acquisition source, to ensure the reproducibility of the experiments.

2. Redundancy between formulas 1…4 and explanation from lines 217-222.

3. Figures 9, 11, 13 contain measured values or values obtained by simulation? If the values were measured, please indicate the measurement equipment used. If the values were obtained by simulation, please give details about the FEM simulation parameters used.

4. In line 298 is assumed that data from fig. 14 were measured. Please indicate the measurement equipment used.

5. In fig. 20 where are a), b) and c)?

6. Please describe how the sensor and excitation structure was moved along the overhead ground wire during the experimental verification.

Reviewer 3 Report

Comments and Suggestions for Authors

In this paper, a device suitable for overhead ground wire defect detection was designed. Some parameters of the device were obtained according simulation. Then, a method based on GA_BP network is proposed to detect the faulty of overhead ground wire.

The detection method proposed in this paper includes software simulation and hardware test, and the experimental results are reasonable and complete, but the following questions need further clarification

1. The abbreviation” GA-BP” needs to be explained. If it is a BP neural network based on genetic algorithm, it is necessary to describe and analyze the optimization process of genetic algorithm.

2. The prediction results of the training data and the test data need to be analyzed separately.

3. The paper states that there are 24 samples, but does not specify the amount of training and testing data collected for neural networks. If there are only 24 data, it is too small for the neural network model, resulting in overfitting or local optimization.

Round 2

Reviewer 1 Report

Comments and Suggestions for Authors

No further comments.

Comments on the Quality of English Language

No further comments.